# AhlX, an *N*-acylhomoserine Lactonase with Unique Properties

**DOI:** 10.3390/md17070387

**Published:** 2019-06-28

**Authors:** Pengfu Liu, Yan Chen, Zongze Shao, Jianwei Chen, Jiequn Wu, Qian Guo, Jiping Shi, Hong Wang, Xiaohe Chu

**Affiliations:** 1Collaborative Innovation Center of Yangtze River DeltaRegion Green Pharmaceuticals, Zhejiang University of Technology, Hangzhou, Zhejiang 310014, China; 2Shanghai Advanced Research Institute, Chinese Academy of Sciences, Pudong, Shanghai 201210, China; 3Key Laboratory of Marine Biogenetic Resources, The Third Institute of Oceanography, Ministry of Natural Resources, Xiamen 361005, China

**Keywords:** quorum sensing, quorum-quenching, *N*-acylhomoserine lactonase, marine, thermostable, salt tolerance

## Abstract

*N*-Acylhomoserine lactonase degrades the lactone ring of *N*-acylhomoserine lactones (AHLs) and has been widely suggested as a promising candidate for use in bacterial disease control. While a number of AHL lactonases have been characterized, none of them has been developed as a commercially available enzymatic product for in vitro AHL quenching due to their low stability. In this study, a highly stable AHL lactonase (AhlX) was identified and isolated from the marine bacterium *Salinicola salaria* MCCC1A01339. AhlX is encoded by a 768-bp gene and has a predicted molecular mass of 29 kDa. The enzyme retained approximately 97% activity after incubating at 25 °C for 12 days and ~100% activity after incubating at 60 °C for 2 h. Furthermore, AhlX exhibited a high salt tolerance, retaining approximately 60% of its activity observed in the presence of 25% NaCl. In addition, an AhlX powder made by an industrial spray-drying process attenuated *Erwinia carotovora* infection. These results suggest that AhlX has great potential for use as an in vitro preventive and therapeutic agent for bacterial diseases.

## 1. Introduction

Bacteria can monitor and respond to their population density using small secreted signaling molecules. The concentration of these molecules increases as the bacteria proliferate. Once it reaches a threshold, the accumulated signal molecules elicit the expression of specialized genes involved in bioluminescence [1], Ti plasmid transfer [2], antibiotic production [3], biofilm formation [4] or pathogenic process [5]. This cell population-dependent behavior is widely known as “quorum sensing (QS)” [6]. The most important QS mechanism identified to date relies on the signaling molecules *N*-acylhomoserine lactones (AHLs), all of which possess the same lactone ring but have different lengths and types of acyl-chains [7]. Most Gram-negative bacteria, including several pathogens such as *Erwinia* sp., *Vibrio* sp., *Yersinia* sp., *Agrobacterium* sp., and *Pseudomonas* sp., use AHLs to modulate their QS behaviors [7].

AHL lactonase catalyzes the hydrolysis of the lactone bond of AHLs and is able to quench AHL-dependent QS signaling. While several bacterial pathogens use QS system to regulate their virulence [2,8], the QS quenching activity of AHL lactonase has always been considered to be an important tool for bacterial disease control. The first AHL lactonase, AiiA, was identified from *Bacillus* sp. 240B1 in 2000 and is comprised of 250 amino acids. AiiA contains a typical “HXHXDH” metallohydrolase family motif and requires zinc ions for its activity. The heterologous expression of AiiA in *Erwinia carotovora* strain SCG1 inhibits the release of its AHL molecules and clearly attenuates its QS-dependent pathogenic processes towards Chinese cabbage, carrots, cauliflower, celery, potatoes, eggplant and tobacco [9]. The transgenic expression of AiiA in plants has been shown to significantly increase the resistance of the plants towards pathogens [10].

At present, apart from the aforementioned organisms, AHL lactonases have also been identified and isolated from *Solibacillus* sp., *Arthrobacter* sp., *Klebsiella* sp., *Rhizobium* sp., *Ochrobactrum* sp., *Microbacterium* sp., *Chryseobacterium* sp., *Rhodococcus* sp., *Oceanicaulis* sp., *Pseudoalteromonas* sp., *Sulfolobus* sp., *Ruegeria* sp., soil metagenomes and human tissues [11,12,13]. However, none of these enzymes have been developed as a commercially available AHL lactonase product for AHL quenching due to their low stability, impeding their further application as in vitro preventive and therapeutic agents for bacterial diseases.

In this study, an AHL lactonase from a marine bacterium, *S. salaries* MCCC1A01339, was identified and characterized. AhlX displays unique properties, including high temperature stability and high salt tolerance. AhlX can be also used to attenuate *E. carotovora* infection. To the best of our knowledge, AhlX is one of the most stable AHL lactonases identified to date and should have excellent potential for further biotechnological use.

## 2. Results

### 2.1. Cloning of the Lactonase-Encoding Gene AhlX

We previously identified a marine bacterium showing clear AHL-degrading activity as indicated in an *A. tumefaciens* NT1 bioassay [14]. This strain was isolated from the Indian Ocean and was initially identified and named as *Halomonas salaria* MCCC1A01339. However, since *Halomonas salaria* has been classified into the *Salinicola* genus and is named as *Salinicola salarius* now [15], we now refer to this strain as *S. salarius* MCCC1A01339. The strain generally requires ≥3% NaCl for a good growth in lab. The reported phylogenetically closest strains to *S. salaries* MCCC1A01339 are *S. salarius* strain M27, as shown by 16S rDNA alignment (>99% identity), and *S. salarius* DSM 18044, as shown by 23S rDNA alignment (>99% identity) (data not shown).

To clone the gene encoding the AHL-degrading enzyme from *S. salarius* MCCC1A01339, we obtained its draft genome sequence via next generation sequencing and subsequent assembly and annotation. From the assembled genome, 3853 genes were identified with an average length of 936 bp. The mean GC content in the gene region is 63.50%. Subsequently, 2688 genes were further annotated. Analysis of the *S. salaries* MCCC1A01339 draft genome enabled us to identify a 768-bp gene (*ahlX*) encoding a potential AHL-degrading enzyme, with the encoded protein sharing the highest identity of 67% to AttM from *A. tumefaciens* [16] and 31% to AiiA from *Bacillus* sp. 240 B1 [9]. AhlX consists of 261 amino acids and has a predicted molecular mass of 29.2 kDa.

### 2.2. Bioinformatic Analysis of AhlX

As mentioned above, AhlX shares relatively low identity with AttM from *A. tumefaciens* [16] and AiiA from *Bacillus* sp. 240 B1 [9]. However, AhlX shares high identity with several predicted AHL lactonases in the database. For example, AhlX shares 92% identity with a predicted AHL lactonase (Gba-pALase) from a bacterium (*Gammaproteobacteria* strain MFB021) isolated from petroleum-contaminated soil [17]; 90% identity with a predicted AHL lactonase (Sso-pAlase) from a bacterium (*Salinicola socius*) isolated from a salt mine (Perm region of Russia) [18]; and 83% identity with a predicted AHL lactonase (Kav-pALase) from a bacterium (*Kushneria avicenniae*) isolated from a salty leaf [19]. 

We further determined the phylogenetic relationship between these AHL lactonases and other well-characterized AHL lactonases. AhlX and the randomly selected predicted AHL lactonases Gba-pALase, Sso-pAlase and Kav-pALaseare were observed to cluster into a small group that is phylogenetically distant from the well-characterized AiiA group from *Bacillus* sp. [9] and AidC from *Chryseobacterium* sp. StRB126 [20]. The phylogenetically closest and well-characterized AHL lactonases to AhlX were AhlK from *Klebsiella pneumoniae* [21] and AttM from *A. tumefaciens* [16] (Figure 1).

The multiple sequence alignment showed that, similar to other well-characterized AHL lactonases, AhlX, Gba-pALase, Sso-pAlase and Kav-pALase all contain the typical zinc binding motif “H^102^XH^104^XD^106^H^107^~H^178^”, which is conserved in the metallo-β-lactamase superfamily [22,23]. Moreover, three additional residues (D^200^, Y^203^ and H^245^), which have been shown to be crucial for the activity of AHL lactonases by interacting with the ligand or through substrate binding, were also observed in AhlX, Gba-pALase, Sso-pAlase and Kav-pALase. Notably, AhlX, Gba-pALase, Sso-pAlase and Kav-pALase harbor five conserved cysteines in their *N*-terminal region, including a C^16^XXC^19^ motif (Figure 2).

While the 3D-structure of several AHL lactonases have been solved, we sought to elucidate the AhlX structure by template based mathematical modeling using the online protein structure and function prediction server I-TASSER (http://zhanglab.ccmb.med.umich.edu/I-TASSER) [24]. As shown in Figure 3a, AhlX contains a αβ/βα sandwich fold, with the helices located in the outer solvent-exposed layer and the β sheets condensed in the core. This structure typically exists in various metallo-β-lactamase family proteins [25]. AhlX harbors 12 strands (β1, residues 5–18; β2, residues 28–41; β3, residues 44–49; β4, residues 72–74; β5, residues 96–100; β6, residues 118–121; β7, residues 154–156; β8, residues 162–166; β9, residues 168–175; β10 residues 182–188; β11 residues 194–198; and β12 residues 240–244) and 10 helices (α1, residues 19–21; α2, residues 65–70; α3, residues 71–88; α4, residues 110–111; α5, residues 123–130; α6, residues 135–136; α7, residues 146–147; α8, residues 205–210; α9, residues 220–236; and α10, residues 247–252). All of the residues (H^102^, H^104^, D^106^, H^107^, H^178^, D^200^, Y^203^ and H^245^) that have been showed to be important in the activity of AHL lactonases are also illustrated and appear to be located on the flexible loop and buried inside the β sheet core, forming an excellent active center (Figure 3b,c). The C^16^XXC^19^ motif is observed to be located close to the active center, like a “door keeper” of the active center (Figure 3b).

### 2.3. Characterization of 3OC8-HSL Degradation by AhlX

To validate the activity of AhlX, *ahlX* was cloned into the vector pET28a and expressed in the host strain *E. coli* BL21 (DE3). The recombinant *E. coli* strain showed clear AHL-degrading activity as detected through the *A. tumefaciens* NT1 bioassay method (Figure 4a). Subsequently, AhlX was purified from the recombinant *E. coli* strain, in which a considerable amount of soluble protein was obtained after induction at 30 °C for 16 h with 0.2 mmol/L IPTG (Figure 4b). AhlX was further purified by one-step Ni^2+^-NTA affinity purification. SDS-PAGE analysis showed that AhlX exhibited a clear band at a molecular mass of approximately 30 kDa, corresponding to its predicted molecular mass (Figure 4b).

To determine whether AhlX works as an AHL lactonase, the substrate 3OC8-HSL was used to test the activity of AhlX, and the resulting degradation product was further analyzed. As shown in Figure 5a, the partial degradation of 3OC8-HSL by AhlX yielded two peaks with the retention times of 4.64 and 3.2 min by HPLC. Since the undegraded 3-OC8-HSL gave a retention time of 4.64 min, the 3.2 min peak is likely a 3OC8-HSL degradation product generated by AhlX. This peak was collected and sent for mass spectrometry analysis. As shown in Figure 5c, the 3.2 min peak collected from HPLC had an *m*/*z* (mass-to-charge ratio) of 258.12, whereas the 3OC8-HSL had an *m*/*z* of 240.12 (Figure 5b). These results indicated that the AhlX-mediated degradation of 3OC8-HSL generated a product with an increased *m*/*z* of 18 (Figure 5d), in line with the property of 3OC8-HS, which is generally produced by AHL lactonase from 3OC8-HSL. These results, in combination with aforementioned bioinformatics analysis, suggest that AhlX is a typical AHL lactonase.

### 2.4. Biochemical Characterization of AhlX

AhlX showed considerable activity at temperatures from 10 to 60 °C, with the optimum activity observed at 40 °C. The enzyme exhibited approximately 50% of its maximal activity at 10 °C and approximately 40% at 60 °C (Figure 6a). AhlX displayed very good activity at pH values of 7.0 to 9.0, but its activity decreased sharply when the pH decreased below 6.0 (Figure 6b). These results suggest that AhlX is sensitive to an acidic catalysis environment.

As shown in Figure 6c, AhlX was activated by several divalent metal ions, including Mg^2+^, Mn^2+^, Co^2+^, Ni^2+^ and Zn^2+^. In the presence of 1 mmol/L Mn^2+^, Co^2+^ and Zn^2+^, the activity of AhlX was increased by over 1-fold, whereas the addition of 1 mmol/L Ni^2+^ increased the activity by over 2-fold. However, Ca^2+^ did not affect the activity of AhlX, and Fe ^2+^ and Cu^2+^ significantly inhibited the activity of AhlX. In addition, 1 mmol/L EDTA also decreased the activity of AhlX by approximately 50%.

AhlX showed activity towards a wide spectrum of substrates. As shown in Figure 6d, AhlX exhibited over 60% activity towards all of the tested AHL-typed substrates, including C4-HSL, C6-HSL, C8-HSL, C10-HSL, C12-HSL, C14-HSL, 3OC6-HSL, 3OC8-HSL, 3OC12-HSL and 3OC14-HSL. Nevertheless, AhlX appeared to perform better against 3-oxo-AHLs and displayed the best activity towards the substrates C6-HSL, 3OC6-HSL and 3OC8-HSL.

Notably, AhlX displayed extraordinary temperature stability. As shown in Figure 7a, after incubation at 0 to 60 °C for 30 min, the activity of AhlX showed nearly no change. In a thermal stability test for an extended period of time, AhlX maintained approximately 97% activity after incubating at 25 °C for 12 days (Figure 7b) and over 40% activity after incubating at 60 °C for 6 h (Figure 7c). Moreover, as shown in Figure 7d, AhlX exhibited strong salt tolerance, with over 80% of the activity of AhlX remaining when 15% NaCl was added into the reaction mixture. Specifically, at the high NaCl concentration of 25%, 60% of AhlX activity was retained.

### 2.5. Quenching the E. carotovora Infection by AhlX

AHL lactonases have been suggested to play important roles in quenching bacterial diseases [9,10,26]. *E. carotovora* SCG1 is a bacterial pathogen that generally elicits the so-called bacterial soft disease in several important vegetables, including potato and cabbage. It has been reported that the infection process of *E. carotovora* SCG1 strongly relies on the expression of pathogenic genes controlled by AHL signaling [27]. To assess the activity of AhlX in the practical control of bacterial diseases, we tested the ability of AhlX to quench *E. carotovora* infection in potato.

*E. coli* is generally used as model organism to test the potential of in vivo function of gene. Therefore, we tested the activity of a recombinant *E. coli* strain in combating *E. carotovora* infection. The *E. coli* strain BL21(DE3) expressing AhlX was co-cultured with *E. carotovora* SCG1 on potato slices. As shown in Figure 8a, after incubating at 30 °C for 40 h, no obvious soft rot symptom was observed on the potato slices inoculated with both strains, whereas clear symptoms were observed on the potato slices inoculated with only *E. carotovora* SCG1 or *E. carotovora* SCG1-*E. coli* BL21 (no AhlX expression). These results suggest that the recombinant *E. coli* strain expressing AhlX inhibited *E. carotovora* infection.

The high temperature stability of AhlX suggested that it should have a potential to be used as an in vitro agent for AHL quenching and bacterial disease control. We further tested the activity of AhlX in vitro against *E. carotovora* infection. To mimic the commercial enzyme production process, we generated AhlX powder directly from the crude lysate of *E. coli* cells expressing AhlX through an industrial spray-drying process. The cell lysate of the *E. coli* culture expressing AhlX was first heated at 60 °C for 30 min to remove the heat-sensitive proteins. Subsequently, a spray-drying process at 80–120 °C was used to generate the dried AhlX powder. The spray-dry process did not significantly alter the activity of AhlX, and the total recovery rate of the AhlX activity from the original lysate supernatant reached 84.06%, suggesting that the AhlX production process was successful (Table 1). As shown in Figure 8b, the spray-dried AhlX powder clearly reduced the *E. carotovora*-induced infectious symptoms of soft rot on potato slices and Chinese cabbage stalks after incubation at 30 °C for 20 h.

## 3. Discussion

In this study, we characterized the lactonase AhlX from the marine bacterium *S. salaries* MCCC1A01339. Members of the *Salinicola* genus are halophilic bacteria. Halophiles grow in the hyper-saline conditions (up to 30% NaCl) and have attracted particular research interest for their ability to resist osmotic stress and salt-induced denaturation. Therefore, these bacteria and their enzymes have great biotechnological potential with respect to environmental remediation, enzyme development and ectoine production [28,29]. To date, QS signaling molecules AHLs have been detected in several halophilic bacteria, including *Chromohalobacter* sp., *Cobetia* sp., *Halomonas* sp., *Halotalea* sp., *Kushneria* sp., *Modicisalibacter* sp. and *Salinicola* sp. [30,31]. However, lactonases, AHL-quenching enzymes, have never been characterized within this group of bacteria. Therefore, the characterization of AhlX should provide a good reference for further physiological and biotechnological studies of AHL quenching by halophilic species.

AhlX shares distinct amino acid sequence with other well-characterized AHL lactonase. AhlX and AHL lactonases to be phylogenetically close, including Gba-pALase, Sso-pAlase and Kav-pALase, all contain five cysteines, residues that are not conserved in other well-characterized AHL lactonases such as AiiA from *Bacillus* sp. 240 B1 and AiiM from *Microbacterium testaceum*. The CXXC motif is suggested to be important in redox sensing-dependent protein function [32]. Despite the detailed roles of these residues being unknown, the conservation of these residues associated with redox-sensing in AhlX, Gba-pALase, Sso-pAlase and Kav-pALase enabled us to envision their different catalytic mechanisms. While it is unclear whether these cysteines are the hallmark of AhlX phylogenetically close group, future studies can help to elucidate their detailed functions.

AhlX retained over 97% activity after incubating at 25 °C for 12 days (288 h) and at 60 °C for 2 h, with even approximately 44% of its activity retained at 60 °C for 6 h. Therefore, AhlX is one of the most stable AHL lactonases identified to date. It has been previously reported that AiiA could maintain >99% activity at 21 °C for 240 h, yet its activity went down sharply after incubating at 45 °C for 2 h [33]. the *N*-acylhomoserine lactonase from the thermophilic bacterium *Geobacillus caldoxylosilyticus* YS-8 was previously shown to retain full activity after pre-incubating at 40 °C for 3 h. However, the activity of this enzyme decreased to approximately 50% of its maximum when incubated at 60 °C for 2 h [34]. Although the mechanism associated with the high stability of AhlX is currently unknown, this property makes it an excellent candidate for use as an in vitro preventive and curative agent for bacterial disease. As shown in Figure 8b, the AhlX powder made through a spray-drying process at 80–120 °C maintained high activity and performed well at preventing *E. carotovora* infection. Further work should focus on testing the activity of AhlX activity against other pathogenic bacteria to boost its commercial use. AhlX showed strong salt tolerance. To the best of our knowledge, AhlX is the first AHL lactonase characterized that can tolerate 25% NaCl. In past decades, pathogenic halophilic bacteria have been identified as the culprits of numerous diseases. For example, *Halomonas* sp. was recently recognized as a human pathogen implicated in infections and contamination in a dialysis center [35]. Since AHL-dependent QS has been implicated in regulating virulence gene expression in many bacterial pathogens, it is easy to envision a connection between AHLs and the pathogenicity of halophiles. Therefore, it is our expectation that future applications of AhlX will involve the quenching of AHLs produced by halophilic bacterial pathogens to control diseases.

AhlX also shares several similar properties with other well-characterized AHL lactonases. AhlX showed optimal activity at pH 7.0–8.0 and 40 °C (Figure 2). These properties resemble those of other reported AHL lactonases, such as AiiA from *Bacillus* sp. 240 B1 [33] and AidC from *Chryseobacterium* sp. strain StRB126 [20]. Notably, AhlX showed broad substrate specificity, with over 60% relative activity observed toward all tested AHLs, with or without 3-oxo substitution (Figure 6d). The previous reports also showed that AiiA from *Bacillus* sp. 240B1 [33] and AiiM from *Microbacterium testaceum* [36] could degrade a wide variety of AHLs. While different bacteria growing in communities tend to use different types of AHLs to control their QS-dependent behavior or survival, the broad substrate specificity of AhlX should make *S. salaries* highly competitive in the niches it occupies.

In summary, in this study, we identified and characterized a highly stable AHL lactonase, AhlX, with a unique amino acid composition. AhlX maintained high activity during its preparation at high temperature using an industrial used spray-drying process, indicating the possibility for its low-cost production at a large scale. The spray-dried AhlX powder demonstrated control of *E. carotovora* infection in vitro, suggesting its potential for the further application in bacterial disease control.

## 4. Materials and Methods

### 4.1. Bacterial Strains and Chemicals

*E. coli* BL21(DE3) was used to express the AhlX protein driven by the pET28a vector. The strain *A. tumefaciens* NT1 expressing an AHL-regulated LacZ was used to evaluate the AHL activity with the bioassay method described by Dong et al. [9]. *E. carotovora* SCG1 was provided by Prof. Ziduo Liu (HuaZhong Agricultural University, Wuhan, China) and used to test the quenching activity of AhlX for bacterial disease.

The AHLs used in this study included *N*-butyryl-l-Homoserine lactone (C4-HSL), *N*-hexanoyl-l-homoserine lactone (C6-HSL), *N*-octanoyl-l-homoserine lactone (C8-HSL), *N*-decanoyl-l-homoserine lactone (C10-HSL), *N*-dodecanoyl-l-homoserine lactone (C12-HSL), *N*-tetradecanoyl-l-homoserine lactone (C14-HSL), *N*-(3-oxohexanoyl)-l-homoserine lactone (3OC6-HSL), *N*-(3-oxooctanoyl)-l-homoserine lactone (3OC8-HSL), *N*-(3-oxododecanoyl)-l-homoserine lactone (3OC12-HSL), and *N*-(3-oxo-tetradecanoyl)-l-homoserine lactone (3OC14-HSL) and were purchased from Cayman Chemical Company (Ann Arbor, MI, USA). Other chemicals, if not specified, were purchased from Sinopharm Group Co. Ltd. (Shanghai, China).

### 4.2. Genome Sequencing and Bioinformatic Analysis

The draft genome sequence of *S. salaries* MCCC1A01339 was obtained using an Illumina MiSeq by Majorbio (Shanghai, China). Gene prediction was performed using Glimmer 3.02 (http://www.cbcb.umd.edu/software/glimmer, the Center for Computational Biology at Johns Hopkins University, Baltimore, MD, USA), and gene annotation was performed by aligning the predicted coding sequences of each gene to those sequences in the non-redundant gene databases (https://www.ncbi.nlm.nih.gov/) and string (http://string-db.org/) and the GO database (http://www.geneontology.org/page/go-database) by Blast analysis. Multiple sequence alignment analysis of AHL lactonases was performed using ClustalX implemented in BioEdit Version 7.0.5 (http://www.mbio.ncsu.edu/bioedit/bioedit.html, Tom Hall; Ibis Therapeutics, Carlsbad, CA, USA) and was illustrated using ESPript Version 3.0 (http://espript.ibcp.fr/ESPript/cgi-bin/ESPript.cgi, “Retroviruses and Structural Biochemistry” research team of the “Molecular Microbiology and Structural Biochemistry” laboratory (UMR5086 CNRS/Lyon University). Lyon Cedex 07, France) [37]. The phylogenetic tree of AHL lactonases was generated using the neighbor-joining method and was illustrated using Tree-view X Version 0.5.0 (http://taxonomy.zoology.gla.ac.uk/rod/treeview.html, Roderic D. M. Page, UK). The 3D-structure modeling of AhlX was performed using the Protein Function and Structure Prediction server, I-TASSER (http://zhanglab.ccmb.med.umich.edu/I-TASSER, Yang Zhang’s Research Group, University of Michigan, Ann Arbor, MI, USA) and was illustrated using Swiss-Pdb Viewer Version 4.1 (Swiss Institute of Bioinformatics, Lausanne, Switzerland).

### 4.3. Expression and Purification of AhlX

To obtain the recombinant AhlX, the *ahlX* gene was PCR amplified using the forward primer GACGTGCATATGGCCGCTCCACGTCTCTATATG and the reverse primer GCTGAATTCTCAAGCGTAGTATTCCGGGGC using standard procedures. The amplified DNA fragment was subsequently ligated into the expression vector pET-28a to generate the plasmid pET28a-*ahlX*, which was subsequently transformed into the *E. coli* DH5α plasmid extraction and sequence analysis. Subsequently, a pET28a-*ahlX* clone with the correct sequence was transformed into the protein expression host *E. coli* BL21(DE3). To express AhlX, 0.2 mmol/L IPTG was added to the culture at an OD_600_ of ~0.6, and the culture was further grown at 30 °C for 16 h. AhlX was purified using Ni^2+^-NTA affinity chromatography following the manufacturer’s protocol (GE Healthcare, Uppsala, Sweden). Gel filtration chromatography was subsequently performed to remove salt ions from the protein sample using Sephadex^TM^ G-25M (GE Healthcare, Uppsala, Sweden). For the detection and analysis of the expressed and purified AhlX, 15% (*v*/*v*) sodium dodecyl sulfate-denatured poly-acrylamide gel electrophoresis (SDS-PAGE) analysis was performed. The purified AhlX protein was stored in −80 °C for later use. Spray-drying of AhlX was performed using a JT-6000Y spray drier (Hangzhou Jtone Electronic Co.Ltd., Hangzhou, China) with an air speed of 100 L/min, an air inlet temperature of 120 °C and an air outlet temperature of 80 °C.

### 4.4. Standard Activity Assay of AhlX

The standard AhlX activity assay was essentially performed as described by Huang et al. [14]. Briefly, 5 μL of purified AhlX was added to 400 μL of 10 mmol/L phosphate buffer (PB, pH 8.0) containing 0.83 mmol/L 3OC8-HSL substrate and incubated at 30 °C for 30 min. The reaction was terminated with SDS solution at a final concentration of 2% and was detected by high-performance liquid chromatography (HPLC) on a Diamonsil^®^C18 column (4.6 × 250 mm, 5 μm, Dikma, Beijing, China) with a mobile phase of C_2_H_3_N:HCOOH: H_2_O(50: 0.2: 49.8) at 1 mL/min. The detection wavelength was set to 201 nm.

### 4.5. Determination of the Mechanism of AHL Degradation by AhlX

To degrade 3OC8-HSL, purified AhlX (final concentration of 18.32 μg/mL) was mixed with 3OC8-HSL (final concentration of 2.0 mmol/L) in 2.0 mL of 10 mmol/L potassium phosphate buffer (pH 8.0) and incubated at 30 °C for 7 h. The reaction mixture was then extracted with an equal volume of ethyl acetate twice, and the product in the organic layer was subsequently collected and concentrated with a vacuum centrifugal concentrator RVC 2-25 CD plus (Christ, Germany). A similar procedure was also used for the reaction mixture without AhlX.

For subsequent chromatographic analysis, each extract was then dissolved in methanol (200 μL), with 8 μL subsequently analyzed by HPLC-MS (Thermo Scientific^TM^ LCQ Fleet^TM^, Thermo Fisher Scientific, Waltham, MA, USA) on a Diamonsil ^®^C18 (250 × 4.6 mm, 5 μm) column with a mobile phase of C_2_H_3_N:HCOOH: H_2_O(50: 0.2: 49.8) at 1 mL/min, a detection wavelength of 201 nm and the column temperature set at 30 °C. Samples ionized by negative electrospray were used for MS analysis with a mass range of *m*/*z* 50 to 500 scanned under the following conditions: sheath gas flow rate of 75 arb, Aux gas flow rate of 20 arb, sweep gas flow rate of 0 arb, I spray voltage of 5 kV, capillary temperature of 300 °C, capillary voltage of −10 V and a tube lens compensation voltage of −125 V.

### 4.6. Biochemical Characterization of AhlX

The optimum temperature and pH for AhlX activity toward 3-OC8-HSL, was assessed at different temperature (10 to 60 °C) and different pH values (KH_2_PO4-K_2_HPO4 buffer for pH 5.0–9.4) under the standard conditions. The effect of metal ions on AhlX activity was evaluated by determining the relative enzyme activity using 3OC8-HSL as substrate in the presence of 1 mmol/L concentrations of different metal ions. To characterize the substrate specificity of AhlX, homoserine lactones with variable chain length and modifications, including C4-HSL, C6-HSL, C8-HSL, C10-HSL, C12-HSL, C14-HSL, 3OC6-HSL, 3OC8-HSL, 3OC12-HSL, and 3OC14-HSL were used as substrates to test its activity. To measure the stability of AhlX at different temperatures, the enzyme was incubated at 0 to 90 °C for 30 min, after which the residual activity was measured under the standard conditions. Furthermore, to measure the time-dependent thermal stability of AhlX, its residual activity was tested after incubation at 60 °C for 0.5, 1, 2, 4, 6, 12 and 24 h, and at 25 °C at 48-h intervals from 0 to 288 h. To determine the salt tolerance of AhlX, 0–25% NaCl was added into the reaction mixture and its residual activity was measured.

### 4.7. Quenching the E. carotovora Infection by AhlX

To test the inhibitory effect of the recombinant strain *E. coli* BL21-pET28a-*ahlX* towards the soft rot infectious disease caused by *E. carotovora*, 7.5 μL of an overnight culture of *E. carotovora* (7.2 × 10^5^ CFU/mL) and 7.5 μL of an overnight culture of *E. coli* BL21-pET28a-*ahlX* (1.8 × 10^6^ CFU/mL) was mixed evenly and inoculated in the center of the potato slices. As a control, 7.5 μL of *E. carotovora* and *E. coli* BL21-pET28a-*ahlX* and 7.5 μL 10 mmol/L PBS (pH 7.4) were mixed with 7.5 μL of PBS and inoculated in the center of the potato slices. After incubating at 30 °C for 40 h, the presence of soft rot symptom was assessed.

To determine the ability of spray-dried AhlX powder to quench the *E. carotovora* infection, 1.5 mg of spray-dried AhlX powder was dissolved into 150 μL PBS buffer and immediately spread onto potato slices and Chinese cabbage stems, after which 1 μL of an overnight culture of *E. carotovora* (3.6 × 10^5^ CFU/mL) was inoculated. The potato slices and Chinese cabbage stems treated with only 10 mmol/L PBS (pH 7.4), spray-dried AhlX (1 g/L) or *E. carotovora* (3.6 × 10^5^ CFU/mL) were used as controls. After incubating at 30 °C for 20 h, the presence of soft rot symptom was assessed.

### 4.8. Nucleotide Sequence Accession Number

The nucleotide sequences of *ahlX* gene, 16S rDNA of *S. salaries* MCCC1A01339, and 23S rDNA of *S. salaries* MCCC1A01339 have been deposited in the GenBank database under the accession numbers KY783591, KY783592 and KY783593, respectively. The draft genome of *S. salaries* MCCC1A01339 was deposited in the BioProject database under the ID PRJNA379806.

## Figures and Tables

**Figure 1 marinedrugs-17-00387-f001:**
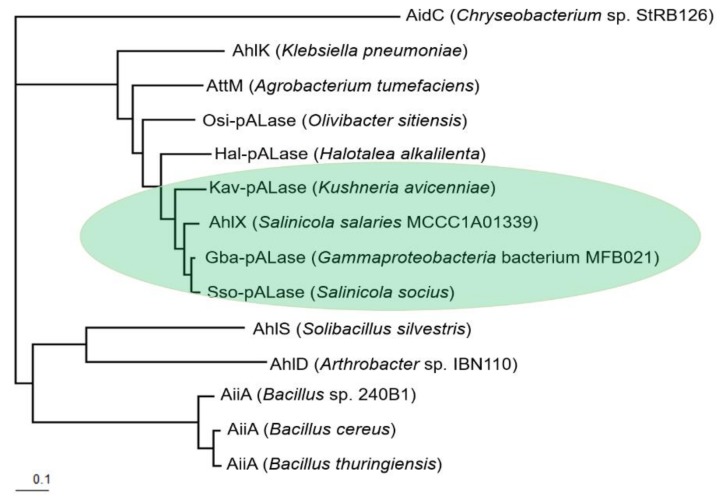
Phylogenetic tree of selected N-acylhomoserine lactone (AHL) lactonases. The AHL lactonase sequences used were AidC (BAM28988.1) from *Chryseobacterium* sp. StRB126; AhlK (AAO47340.1) from *K. pneumoniae*; AttM (AAL13075.1) from *A. tumefaciens*; Osi-pALase (predicted AHL lactonase; WP_028296339.1) from *Olivibacter sitiensis*; Hal-pALase (predicted AHL lactonase; WP_064123850.1) from *Halotalea alkalilenta*; Kav-pALase (SFC91482.1) from *Kushneria avicenniae*; AhlX from *S. salaries* MCCC1A10339; Gba-pALase (WP_035471581.1) from *Gammaproteobacteria* bacterium MFB021; Sso-pALase (WP_075571152.1) from *Salinicola socius*; AhlS (BAK54003.1) from *Solibacillus silvestris*; AhlD (AAP57766.1) from *Arthrobacter* sp. IBN110; AiiA (AAF62398.1) from *Bacillus* sp. 240B1;AiiA (AAL98724.1) from *Bacillus cereus* and AiiA (AAL98718.1) from *Bacillus thuringiensis*. The AHL lactonases from halophilic bacteria are highlighted in light blue.

**Figure 2 marinedrugs-17-00387-f002:**
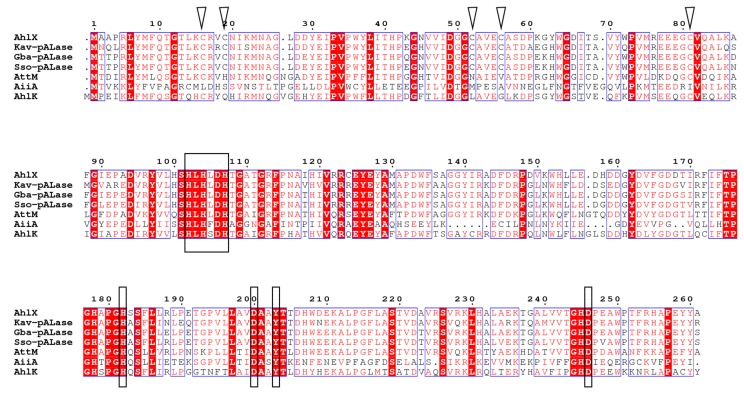
Multiple sequence alignment between AhlX and other AHL lactonases. AhlX, AHL lactonase from *S. salaries* A10339; Kav-pALase (SFC91482.1), predicted AHL lactonase from the halophilic bacterium *K. avicenniae*; Gba-pALase (WP_035471581.1), predicted AHL lactonase from the halophilic bacterium *Gammaproteobacteria* bacterium MFB021; Sso-pALase (WP_075571152.1), predicted AHL lactonase from the halophilic bacterium *S. socius*; AttM (AAL13075.1), AHL lactonase from *A. tumefaciens*; AiiA (AAF62398.1), AHL lactonase from *Bacillus* sp. 240 B1; and AhlK (AAO47340.1), AHL lactonase from *K. pneumoniae*. The conserved residues that are crucial for the activity of AHL lactonases are indicated with black rectangles. The conserved cystines present in Kav-pALase, AhlX, Gba-pALase and Sso-pALase are indicated by black triangles.

**Figure 3 marinedrugs-17-00387-f003:**
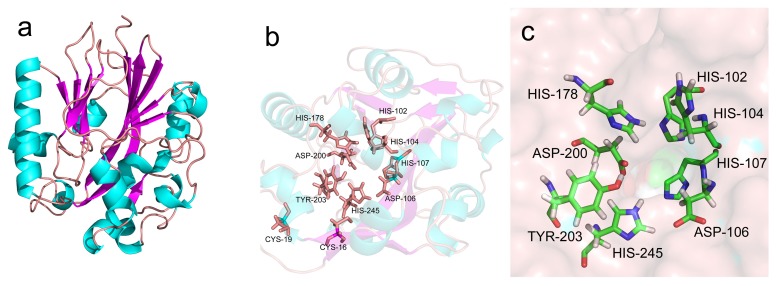
In silico modeling of the structure of AhlX. The 3D-structure modeling of AhlX was performed using the Protein Function and Structure Prediction server, I-TASSER (http://zhanglab.ccmb.med.umich.edu/I-TASSER) [24] and illustrated using Swiss-PdbViewer. (**a**) The modeled 3D structure of AhlX; and (**b**) predicted functional sites in AhlX. Previously, identified important active site residues of AHL lactonases (H^102^, H^104^, D^106^, H^107^, H^178^, D^200^, Y^203^ and H^245^) and the distinct C^16^XXC^19^ motif identified in this study are illustrated. (**c**) A partial enlarged view of predicted functional sites in AhlX. The protein was shown as surface model, and catalytic residues were shown as stick model, respectively.

**Figure 4 marinedrugs-17-00387-f004:**
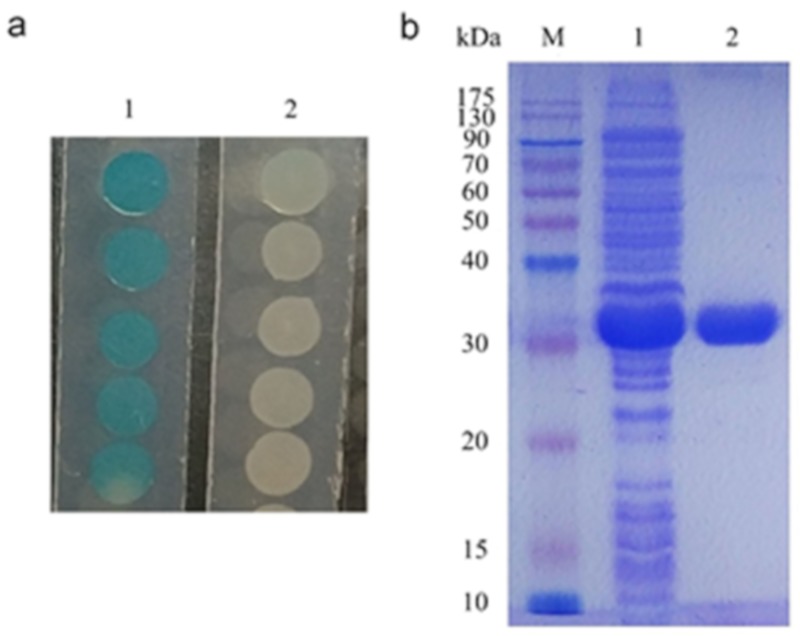
Degradation of 3OC8-HSL by AhlX and purification of recombinant AhlX. (**a**) Degradation of 3OC8-HSL by *E. coli* BL21-pET28a-*ahlX.* Lane 1, 0.5 μmol/L of the 3OC8-HSL sample; Lane 2, 3OC8-HSL degradation product generated by AhlX; (**b**) SDS-PAGE analysis of AhlX expressed in *E. coli* BL21 (DE3) and purified recombinant AhlX. Lane M, protein marker; Lane 1, supernatant of induced *E. coli* cells harboring pET28a-*ahlX*; Lane 2, purified recombinant AhlX.

**Figure 5 marinedrugs-17-00387-f005:**
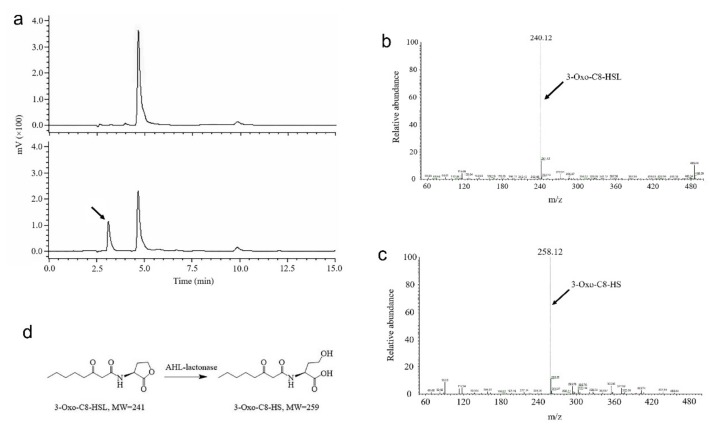
HPLC and LC-MS analysis of 3OC8-HSL the degradation product generated by AhlX. (**a**) HPLC analysis of 3OC8-HSL and its degradation product generated by AhlX. The upper panel shows the HPLC profile of 3OC8-HSL, while the lower panel shows the HPLC profile with the 3OC8-HSL degradation product generated by AhlX. The arrow indicates the expected degradation product at a retention time of 3.2 min. (**b**) LC-MS profile of 3OC8-HSL. The arrow indicates 3OC8-HSL with an *m*/*z* of 240.12. (**c**) LC-MS analysis of the 3OC8-HSL degradation product generated by AhlX. The arrow indicates the expected degradation product with an *m*/*z* of 258.12. (**d**) Schematic illustration of the mechanism of 3OC8-HSL degradation by AHL lactonase.

**Figure 6 marinedrugs-17-00387-f006:**
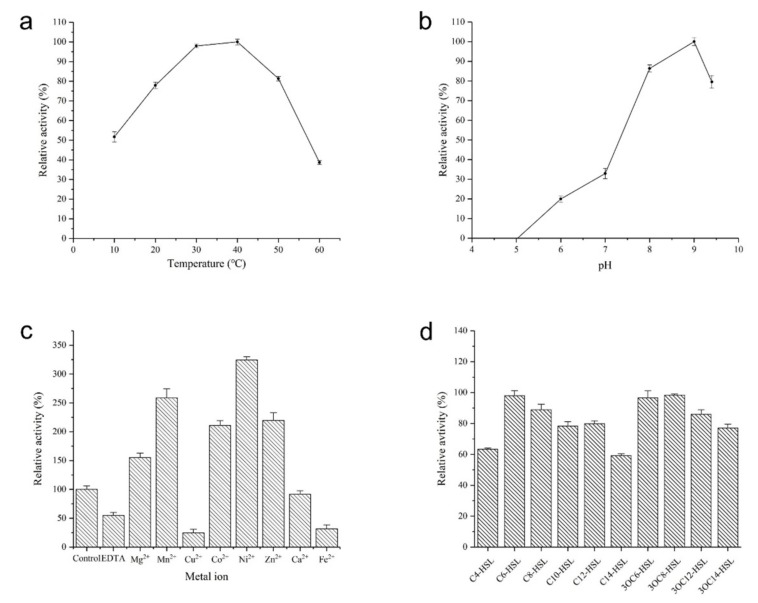
Biochemical characterization of AhlX. (**a**) Effects of temperature on the activity of AhlX. (**b**) Effects of pH on the activity of AhlX. (**c**) Effects of different divalent cations on the activity of AhlX. (**d**) Substrate specificity of AhlX.

**Figure 7 marinedrugs-17-00387-f007:**
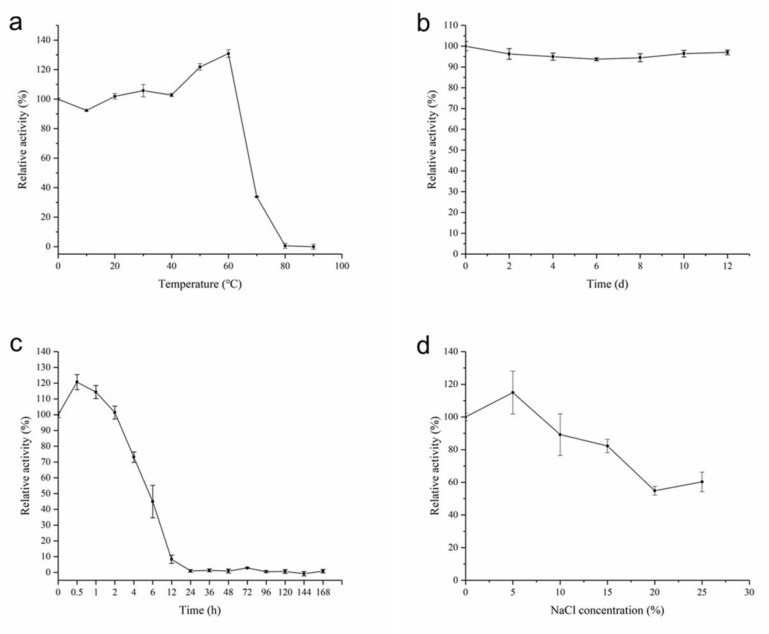
The temperature stability and salt tolerance of AhlX. (**a**) The thermal stability of AhlX at different temperatures. (**b**) The time-dependent thermal stability of AhlX at 25 °C. (**c**) The time-dependent thermal stability of AhlX at 60 °C. (**d**) Effect of different NaCl concentrations on the activity of AhlX.

**Figure 8 marinedrugs-17-00387-f008:**
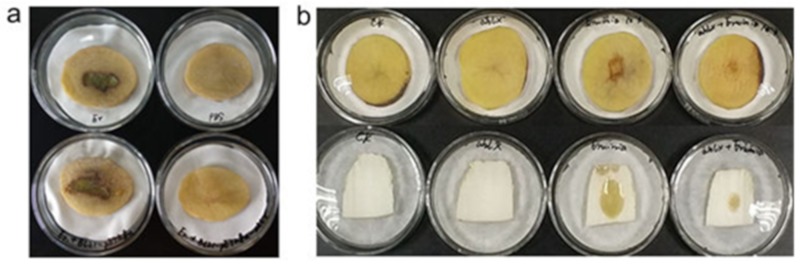
The inhibitory effect of AhlX toward the soft rot infectious disease caused by *E. carotovora* SCG1. (**a**) The inhibitory effect of *E. coli* BL21-pET28a-*ahlX* towards the soft rot infectious disease caused by *E. carotovora* SCG1. The surface-sterilized potato slices were inoculated separately with *E. carotovora* SCG1 (upper left), Phosphate Buffered Saline (PBS) (pH = 7.4) (upper right), the *E. carotovora* SCG1 and *E. coli* BL21-pET28a mixture (bottom left), or the *E. carotovora* SCG1 and *E. coli* BL21-pET28a-*ahlX* mixture (bottom right). (**b**) The inhibitory effect of spray-dried AhlX on soft rot caused by *E. carotovora* SCG1. The surface-sterilized potato slices and Chinese cabbage stalks were treated separately with PBS (pH = 7.4), spray-dried AhlX, *E. carotovora* SCG1, *E. carotovora* SCG1 and spray-dried AhlX (from left to right).

**Table 1 marinedrugs-17-00387-t001:** The activity of AhlX prepared by different methods.

Purification Step	Total Protein (mg)	Specific Activity (Units/mg)	Total Activity (Units)	Enzyme Activity Recovery (%)
Lysate supernatant	507.06	108.89	5.52 × 10^4^	100
Heat treatment	247.35	223.76	5.53 × 10^4^	100.24
Heat treatment + Spray drying	203.36	228.25	4.64 × 10^4^	84.06

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
