# Peer review of "AhlX, an N-acylhomoserine Lactonase with Unique Properties"

_marinedrugs, 2019, doi:10.3390/md17070387_

Round 1

Reviewer 1 Report

Nice work! Appropriate experiments, clearly presented--attached copy with few wording options and typos.

Author Response

Dear reviewer,

We have considered all the comments provided by editors and reviewers carefully. According to the suggestion, some needed modification of the manuscript has also been done. The attachment is a letter stating our responses to each of the reviewers' suggestions. All explanations have been arrayed item by item according to the order of comments provided by reviewers, with a label indicating the page and line number of the revised text. We have highlighted all the sections that have been changed in the revised text. We hope these modifications would be appropriate.

I look forward to hearing from you.

Thank you very much.

Yours sincerely,

Xiaohe Chu

Responses to Reviewer1: (in red font)

Reviewers' comments:
Reviewer #1:

Point 1: L26: "anindustrialspray-drying"; L27: "in vitro" italics

Response 1: This revision has been done in the manuscript.

Point 2: L212-213

Response 2: This revision has been done in the manuscript.

Point 3: L299

Response 3: This revision has been done in the manuscript.

Point 4: L362-366

Response 4: The revisions have been done in the manuscript.

Reviewer 2 Report

Dear Authors,

a valuable finding reported in a straight way. However, the presentation is week - as i miss the actual need to be discussed and overall there are many small issues showing that this draft need to be finalized prior submitting. Further, I do not see the why it fits to this journal - so you need to makre this point more clear!

general:

"N-" italics

"in vitro" italics

a number of space issues at the end of sentences or next to lists or abbreviations etc....

detailed comments:

L26: correct "anindustrialspray-drying"; L48: correct "tomaintainits" ... and so on "aforementioned"

L56 formatting issues

L88: correct "Gammaproteo bacteria"

Figure 2 need to be presented in a reader friendly form - i suggest u use another program for visualization.

Figure 3; please add a zoom in into the active site as part c; and in general the qualitiy could be improved by having a higher resolution by means of pymol and using there the pixel based zoom.

L149: correct; A. Tumefacies

Fig5a indicate peaks by adding a legend

metal sensitivity and metal activity; as the protein was obtained from Ni chromatography this test need to outlined in more detail; have metal ions from the purification removed, how was Zn2+ verified as the natural ion present and was it removed prior supplementing with other metal ions ... if not it is a mix of different enzyme states most likely.

Author Response

Dear reviewer,

We have considered all the comments provided by editors and reviewers carefully. According to the suggestion, some needed modification of the manuscript has also been done. The attachment is a letter stating our responses to each of the reviewers' suggestions. All explanations have been arrayed item by item according to the order of comments provided by reviewers, with a label indicating the page and line number of the revised text. We have highlighted all the sections that have been changed in the revised text. We hope these modifications would be appropriate.

I look forward to hearing from you.

Thank you very much.

Yours sincerely,

Xiaohe Chu

Responses to Reviewer and Editor: (in red font)

Reviewers' comments:
Reviewer #2:

Point 1: a valuable finding reported in a straight way. However, the presentation is week - as i miss the actual need to be discussed and overall there are many small issues showing that this draft need to be finalized prior submitting. Further, I do not see the why it fits to this journal - so you need to makre this point more clear!

Response 1:Many small issues have been revised and the language and grammar of manuscript has been modified by language editing service at American Journal Experts.

Point 2:Further, I do not see the why it fits to this journal - so you need to makre this point more clear!

Response 2: We really appreciate the command from reviewer. Currently, we can see that, owing to the grand prospect of protein and peptide drugs,a number of studies have been placed on the exploitation of these drugs. As described in our main text, AhlX, derived from the marine isolate, Salinicola salaria MCCC1A01339, can quench the signal molecule N-acylhomoserine lactone (AHL), one key molecule that govern the infection process of a number of pathogenic bacteria. We do think AhlX, as a marine bacterium derived enzyme, can be used as a novel biocontrol agent or protein drug in biological control of bacterial diseases. This should fit well to the scope of Marine Drugs. Indeed, we have seen similar “AHL lactonases” research published in Marine Drugs before (see reference 1 and 2).

Point 3: "N-" italics

Response 3: This revision has been done in the manuscript.

Point 4:"in vitro" italics

Response 4: This revision has been done in the manuscript.

Point 5: a number of space issues at the end of sentences or next to lists or abbreviations etc....

Response 5: This revision has been done in the manuscript.

Point 6: L26: correct "anindustrialspray-drying"; L48: correct "tomaintainits" ... and so on "aforementioned"

Response 6: This revision has been done in the manuscript.

Point 7: L56 formatting issues

Response 7: This revision has been done, see L60.

Point 8: L88: correct "Gammaproteo bacteria"

Response 8: This revision has been done, see L97.

Point 9: Figure 2 need to be presented in a reader friendly form - i suggest u use another program for visualization.

Response 9: This revision has been done, see Figure 2.

Point 10:Figure 3; please add a zoom in into the active site as part c; and in general the qualitiy could be improved by having a higher resolution by means of pymol and using there the pixel based zoom.

Response 10: This revision has been done, see Figure 3.

Point 11:L149: correct; A. Tumefacies

Response 11: This revision has been done, see L165 and L68.

Point 12:Fig5a indicate peaks by adding a legend 

Response 12: This revision has been done, see Fig5a.

Point 13:metal sensitivity and metal activity; as the protein was obtained from Ni chromatography this test need to outlined in more detail; have metal ions from the purification removed, how was Zn2+ verified as the natural ion present and was it removed prior supplementing with other metal ions ... if not it is a mix of different enzyme states most likely.

Reponse 13

This is a very good question! We have added several sentences in the main-text regarding the protein purification hopefully to explain this experiment better (see line378). Regarding the protein purification, an affinity chromatography and dialysis process were employed in order to generate the pure AhlX and stored it in the PBS buffer. From this, we think, apart from Na+ of the PBS buffer and AhlX (possibly) strongly-bound metal ions, other metal ions should be all removed from the protein. There might be trace amount of different metal ions associated with purified AhlX even after purification process, resulting a “mix” enzyme state, but this should not affect our research aim and is beyond our concern.

Indeed, the aim of this metal sensitivity test is to study on the effect of different external metal ions on the activity of AhlX. Similar experiment (using similar way) has also been done in other AHL lactonase research, see reference 3-5 below. The activity of one protein can be affected by different metal ions because of its amino acid motif, secondary or higher structure…and et.c. Therefore, to some extent, this metal sensisitivty test reflects the interaction between the external metal ions and the protein. On one hand, it reflects the biochemical property of the enzyme; on the other hand it gives information about how to use the enzyme (regarding metal ions feeding).

Hopefully, this answer the reviewer’s question. If not, we welcome more discussion and can give more explanation.

References

1. Cai X.; Yu M.; Shan H.; Tian X.; Zheng Y.; Xue C.; Zhang X. Characterization of a Novel N-Acylhomoserine Lactonase RmmL from Ruegeria mobilis YJ3. Mar. Drugs 2018, 16, 370

2. Zhang J.; Wang J.; Feng T.; Du R.; Tian X.; Wang Y.; Zhang X. Heterologous Expression of the Marine-Derived Quorum Quenching Enzyme MomL Can Expand the Antibacterial Spectrum of Bacillus brevis. Mar. Drugs 2019, 17.

3. Dong, W.; Zhu, J.; Guo, X.; Kong, D.; Zhang, Q.; Zhou, Y.; Liu, X.; Zhao, S.; Ruan, Z., Characterization of AiiK, an AHL lactonase, from Kurthia huakui LAM0618T and its application in quorum quenching on Pseudomonas aeruginosa PAO1. Scientific Reports 2018; Vol. 8.

4. Tang, K.; Su, Y.; Brackman, G.; Cui, F.; Zhang, Y.; Shi, X.; Coenye, T.; Zhang, X.-H.; Parales, R. E., MomL, a Novel Marine-DerivedN-Acyl Homoserine Lactonase from Muricauda olearia. Appl. Environ. Microbiol.  2015, 81, (2), 774-782.

5. Wang, L. H.; Weng, L. X.; Dong, Y. H.; Zhang, L. H., Specificity and enzyme kinetics of the quorum-quenching N-Acyl homoserine lactone lactonase (AHL-lactonase). J Biol Chem 2004, 279, (14), 13645-51

Round 2

Reviewer 2 Report

Dear Authors,

it improved significantly!

only minor points:

Figure 1; the resolution seems a bit low; "sp." should not be italics; "Gammaproteobacteria" should be italics.

Figure 2; triangles are black and not purple?

methods; L 369, 370, ... - delete trademark signs

Author Response

Dear reviewer,

We have considered your comments carefully. According to the suggestion, some needed modification of the manuscript has also been done. The attachment is a letter stating our responses to each of the reviewers' suggestions. All explanations have been arrayed item by item according to the order of comments provided by reviewers, with a label indicating the page and line number of the revised text. We have highlighted all the sections that have been changed in the revised text. We hope these modifications would be appropriate.

I look forward to hearing from you.

Thank you very much.

Yours sincerely,

Xiaohe Chu

Responses to Reviewer: (in red font)

Reviewers' comments:
Reviewer #2:

Point 1: Figure 1; the resolution seems a bit low; "sp." should not be italics; "Gammaproteobacteria" should be italics.

Response 1: Some revisions have been done in the manuscript. see Figure 1.

Point 2: Figure 2; triangles are black and not purple?

Response 2: This revision has been done, see L143.

Point 3: methods; L 369, 370, ... - delete trademark signs

Response 3: This revision has been done in the manuscript, see L372 and 373.